

# A redundancy-removing feature selection algorithm for nominal data

Zhihua Li[1,2,3,4] and Wenqu Gu[2,3]

[1] Key Laboratory of Advanced Process Control for Light Industry Ministry of Education, JiangSu, China
[2] Engineering Research Center of Internet of Things Technology, Application Ministry of Education, JiangSu, China
[3] Department of Computer Science, Engineering School of Internet of Things Engineering, Jiangnan University, JiangSu, China
[4] Department of Computer Science, Georgia State University, Atlanta, GA, United States of America

## ABSTRACT

No order correlation or similarity metric exists in nominal data, and there will always be more redundancy in a nominal dataset, which means that an efficient mutual information-based nominal-data feature selection method is relatively difficult to find. In this paper, a nominal-data feature selection method based on mutual information without data transformation, called the redundancy-removing more relevance less redundancy algorithm, is proposed. By forming several new information-related definitions and the corresponding computational methods, the proposed method can compute the information-related amount of nominal data directly. Furthermore, by creating a new evaluation function that considers both the relevance and the redundancy globally, the new feature selection method can evaluate the importance of each nominal-data feature. Although the presented feature selection method takes commonly used MIFS-like forms, it is capable of handling high-dimensional datasets without expensive computations. We perform extensive experimental comparisons of the proposed algorithm and other methods using three benchmarking nominal datasets with two different classifiers. The experimental results demonstrate the average advantage of the presented algorithm over the well-known NMIFS algorithm in terms of the feature selection and classification accuracy, which indicates that the proposed method has a promising performance.

Corresponding author
Zhihua Li, zhli@jiangnan.edu.cn

## INTRODUCTION

There are two main feature reduction approaches in data analysis, feature extraction and feature selection (*Jain, Duin & Mao, 2000*). Feature extraction aims at creating new features that are based on transformations or combinations of the raw feature set, and feature selection means selecting one group of the most efficient features from a certain dataset according to certain evaluations that are based on the goodness of a feature, with the purpose of decreasing the feature dimensionality (*Jain, Duin & Mao, 2000*; *Tesmer & Estévez, 2004*; *John, Kohavi & Pfleger, 1994*). This approach is one of the major methods

of feature reduction for high-dimensional data. The evaluation basis includes (*Tesmer & Estévez, 2004*) various distance measurements (*Kira & Rendel, 1992*), a dependency measurement (*Modrzejew, 1993*), a consistency measurement (*Almuallim & Dietterich, 1991*), a probability density measurement (*Battiti, 1994*), a rough set measurement (*Hu, Xie & Yu, 2007*), an information measurement (*Kwak & Ch, 2002*; *Kwak & Choi, 2002*; *Torkkola, 2003*) and some other derivations such as based on optimization strategies (*Hou, Nie & Li, 2014*). Regardless of which evaluation basis is taken, the goal is to keep the number of selected features as small as possible, to avoid increasing the computational cost of the learning algorithm as well as the classifier complexity (*Tesmer & Estévez, 2004*). There have been numerous studies in the literature about various feature selection algorithms that depend on different evaluation bases.

Among them, the information theory-based feature selection algorithm that operates with respect to the selected features and raw dataset can involve less work while processing the data after the optimization transformation and the maximization of the mutual information (MI) between class labels. The mutual information-based feature selection MIFS (*Battiti, 1994*) algorithm is based on this basis, which utilizes greedy selection to guarantee that candidate features of the evaluation function can satisfy the final effective features. After studying the cases of unbalance of the evaluation function in the MIFS algorithm, the MIFS-U (*Kwak & Choi, 2002*) algorithm was proposed. The adaptive feature selection criterion was studied, and then the AMIFS (*Tesmer & Estévez, 2004*) algorithm was presented to cater to feature selection in high-dimensional data. Considering the max-dependency and min-redundancy as a whole, the mRMR (*Peng, Long & Ding, 2005*) algorithm was given. By reconstructing the evaluation function, the NMIFS (*Estévez, Tesmer & Perez, 2009*) algorithm was then proposed. Both the mRMR and NMIFS algorithms can distinctly decrease the redundancy in the selected features.

However, these algorithms also have their disadvantages. For example, MIFS (*Battiti, 1994*) and MIFS-U (*Kwak & Choi, 2002*) fail to consider the mutual information between the candidate features with selected subsets and class labels as well as the influences of the classification results. According to the aforementioned algorithms, their approximation computing of MI can perform only continuous-attributed data.

Nominal data exist in a broad range of practical applications. This kind of data is typically characterized by having no order information, being discrete, and having semantics (*Chow, Wang & Ma, 2008*). No similarity metric or order correlation (*Chow, Wang & Ma, 2008*; *Tang & Mao, 2005*; *Minho & Ramakrishna, 2009*) exists in nominal data. The "distance" in pattern recognition is hard to identify in it, which makes the measurement of the similarity or dissimilarity difficult. Given the existing characteristics of nominal data, some problems in the feature selection appear. Due to having non-order information and discrete and non-metric data distributions, the features of different classes even intersect with one another (*Chow, Wang & Ma, 2008*; *Li, Yang & Gu, 2013*; *Minho & Ramakrishna, 2009*). Thus, most well-known feature selection algorithms are unsuitable for nominal-data feature selection or nominal-data feature extraction.

Considering the above disadvantages specifically and aiming at the nominal-data feature selection and its specificity, this paper presents the new scheme of MI-based More Relevance Less Redundancy (MRLR) through the redefinition of the features' information amounts, the relevance degree between the features and the conditional MI as well as re-construction of the corresponding new approximation computation method for MI with respect to nominal data. On the other hand, through studying the evaluation function and specifically the insufficient consideration of redundancy between features in most evaluation functions, this paper also creates a new evaluation function for nominal data. The new evaluation function not only considers the correlation between features and class labels but also accounts for the mutual correlation between features. In this way, the computation of MI for nominal-data features can be solved, and at the same time, an overly high redundancy of the selected subset caused by the redundant features can also be overcome. Combining the innovations together into a new method, the Redundancy-removing MRLR RedremovingMRLR algorithm is proposed.

Several experiments were arranged. A total of three benchmarking nominal datasets are employed to compare the effectiveness and the efficiency by the naive Bayes classifier and the decision tree classifier on the selected subsets of RedremovingMRLR, MRLR (*Gu & Li, 2013*) and NMIFS algorithm. The experimental results show that the newly proposed scheme can deliver promising results.

The remainder of this paper is organized as follows. The related work and the new definitions are introduced in 'Notation and Related Studies.' In 'The proposed algorithms,' we derive the framework of the proposed feature selection algorithm RedremovingMRLR. Promising experimental results on benchmarking datasets are presented in 'Results and discussion,' which are followed by the concluding remarks in 'Conclusions.'

## NOTATION AND RELATED STUDIES

The related studies are introduced in 'Related work,' and some new definitions and necessary terminology are presented in 'Notation and Definitions.'

### Related work

In this paper, we also use MI, which addresses taking the MI as a matrix of relevance and redundancy among the features, to study the nominal-data feature selection methods. Some of the literature about feature selection methods that are based on MI have been issued, and References (*Tesmer & Estévez, 2004*; *Battiti, 1994*; *Kwak & Choi, 2002*; *Peng, Long & Ding, 2005*; *Estévez, Tesmer & Perez, 2009*; *Chow, Wang & Ma, 2008*) are benchmarks.

MIFS (*Battiti, 1994*) selects the features that maximize the information about the classes, which are corrected by subtracting a quantity that is proportional to the average MI of the previously selected features. When there are many irrelevant and redundant features, the performance of MIFS degrades because it penalizes too much redundancy.

MIFS-U (*Kwak & Choi, 2002*) proposed an enhancement of the MIFS algorithm that makes a better estimation of the MI between the input features and output classes. However, although MIFS-U is usually better than MIFS, its performance also degrades in the presence of many irrelevant and redundant features (*Estévez, Tesmer & Perez, 2009*).

AMIFS (*Tesmer & Estévez, 2004*) presents an enhancement of MIFS and MIFS-U that overcomes their limitations in high-dimensional feature selection. An adaptive selection criterion is proposed in such a way that the trade-off between discarding the redundancy or irrelevance is adaptively controlled (*Estévez, Tesmer & Perez, 2009*), which eliminates the need for a user-predefined parameter, i.e., a fixed parameter.

By deriving an equivalent form called the minimal-redundancy-maximal-relevance criterion, for first-order incremental feature selection, mRMR (*Peng, Long & Ding, 2005*) proposes a framework to minimize the redundancy and uses a series of intuitive measures of relevance and redundancy to select the most promising features. The mRMR algorithm combines other wrapper feature selection methods to select good features first according to the maximal statistical dependency criterion based on MI. It can select promising features for both continuous and discrete datasets (*Peng, Long & Ding, 2005*; *Estévez, Tesmer & Perez, 2009*).

NMIFS (*Estévez, Tesmer & Perez, 2009*) takes the average normalized MI as a measure of the redundancy among the features (*Tesmer & Estévez, 2004*); it is also an enhancement of the MIFS, MIFS-U, and mRMR methods. NMIFS outperforms the MIFS, MIFS-U, and mRMR methods without requiring a predefined parameter.

However, UFSN (*Chow, Wang & Ma, 2008*) can directly handle the nominal-data feature selection and overcome the shortcomings of converting data from nominal into binary. Nevertheless, UFSN must depend on the cluster algorithm at the beginning, and it has a high complexity for large datasets during the process of clustering.

This paper focuses on three issues that have not been covered in earlier work and highlights them as follows. First, by rewriting the evaluation function in NMIFS (*Estévez, Tesmer & Perez, 2009*); thus, the new presented algorithm gives a stronger penalty factor to the redundancy than that in NMIFS. Second, the new algorithm considers the redundancy and relevance of the features as a whole, which the NMIFS does not. The new proposed algorithm also realizes less redundancy and more relevance regardless of the relationships between the features, as well as between the features and the class labels. Third, by aiming at the feature selection for nominal data by MI measurement and simplifying the computation of MI between the features of nominal data, several new definitions are given.The experimental results show that the RedremovingMRLR algorithm is effective in nominal-data feature selection.

## Notation and definitions

To realize the nominal-data feature selection efficiently, several new definitions and the corresponding computing methods for them are first given, as follows:

**Definition 1** Given n values of the *i*th feature $f_i$, i.e., $\{a_1, a_2, \ldots, a_n\} \in f_i$, then the information amount of $f_i$ can be expressed as

$$I(f_i) = -\sum_{i=1}^{n} p_i \log_2(p_i) \tag{1}$$

where $p_i$ represents the frequency of each value in $f_i$, namely, $p_i = \frac{a_i}{|f_i|}$.

**Definition 2** Conditional mutual information between two different features $f_i$ and $f_j$ can be expressed as

$$E(f_i; f_j) = -\sum_{j=1}^{m} p_j I(f_{ij}) \tag{2}$$

where $E(f_i; f_j)$ represents the dependence degree of the $i$th feature $f_i$ on the $j$th feature $f_j$, where $f_j$ is determined. Here, $m$ denotes the number of values in $f_j$.

**Definition 3** According to the above definition, the relevance degree between $f_i$ and $f_j$ can be expressed as

$$G(f_i; f_j) = I(f_i) - E(f_j; f_i) = I(f_j) - E(f_i; f_j). \tag{3}$$

It can be obtained that the relevance degree between $f_i$ and $f_j$ satisfies symmetry. $I(f_i)$ can be obtained by Formula (1), and $E(f_j; f_i)$ can be obtained by Formula (2).

**Definition 4** The preliminary evaluation function of the feature selection for nominal data on each $f_i$ can be expressed as

$$J(f_i) = G(S \cup f_i; C) - \left(\frac{1}{|S|}\right) \sum_{f_s \in S} G(f_s; f_i) \tag{4}$$

where $G(S \cup f_i; C)$ represents the relevance degree between the class labels $\boldsymbol{C}$ and the selected subset $\boldsymbol{S}$ after being added to the candidate feature $f_i$. Similarly, the penalty factor $\beta$ is a user-predefined parameter in MIFS and MIFS-U, and it is difficult to determine. To overcome this limitation, this paper here replaces it with $1/|S|$.

# THE PROPOSED ALGORITHMS

Considering the specificity of nominal-data feature selection, this paper performs the following research:

## Basic idea of the algorithms

Based on the above, the algorithm should select the features via a maximum MI with class labels. More concretely, the algorithm should also consider the MI between different features to avoid overlarge feature redundancy. In this way, the uncertainty of the other features can be determined to the maximum extent. This feature selection algorithm first chose the feature that had the largest relevance to the class labels. Then, the relevance degrees between the candidate feature and the selected features as well as the class labels are computed. Finally, the feature that has more relevance to the class label and less redundancy with the selected features is selected. After several iterations, the selected subset that satisfies the conditions can be obtained.

## Redundancy-removing feature selection algorithm

Inspired by NMIFS (*Estévez, Tesmer & Perez, 2009*), the MI between the candidate feature $f_i$ in the dataset and the selected feature $f_s$ in subset $S$ is shown in Formula (5).

$$MI(f_i; f_s) = H(f_i) - H(f_i|f_s) = H(f_s) - H(f_s|f_i) \tag{5}$$

where $H(f_i)$ and $H(f_s)$ represent the entropy of the feature. $H(f_i|f_s)$ and $H(f_s|f_i)$ represent the corresponding conditional entropies.

From Formula (5), it can be known that Formula (6) is satisfied.

$$0 \leq MI(f_i, f_s) \leq \min\{H(f_i), H(f_s)\}. \tag{6}$$

Furthermore, the concept of a redundancy evaluation operator (*Peng, Long & Ding, 2005*; *Estévez, Tesmer & Perez, 2009*) is introduced here and is shown in Formula (7). This approach aims to evaluate the redundancy degree between the features $f_s$.

$$NMI(f_i; f_s) = \frac{MI(f_i; f_s)}{\min\{H(f_i), H(f_s)\}} \tag{7}$$

where in Formula (7), it can be known that $NMI(f_i; f_s) \in [0, 1]$. When $NMI(f_i; f_s) = 0$, the two features are mutually independent, whereas $NMI(f_i; f_s) = 1$ means that there is great redundancy between the candidate feature $f_i$ and the selected features.

On the basis of MIFS (*Battiti, 1994*), NMIFS (*Estévez, Tesmer & Perez, 2009*) evolved a new strategy of a redundancy matrix operator for the evaluation function for feature selection and then proposed the NMIFS algorithm (*Estévez, Tesmer & Perez, 2009*), which selected the candidate feature $f_i$ with a maximum evaluation function value as the preferred feature and added it into the selected subset **S**. The greatest contribution of the NMIFS algorithm is that it addresses automatically preventing the feature that has more redundancy with the selected subset in the selection process (*Estévez, Tesmer & Perez, 2009*). However, the NMIFS algorithm can adapt only to continuous-attributed data instead of nominal data. Therefore, inspired by the redundancy-removing idea in NMIFS algorithm, Formula (4) is modified into Formula (8).

$$J(f_i) = G(S \cup f_i; C) - \left(\frac{1}{|S|}\right) \sum_{f_s \in S} \frac{G(f_s; f_i)}{\min\{H(f_i), H(f_s)\}}. \tag{8}$$

It is clear that the redundancy between the candidate features and selected features is guided by Formula (8). Thus, Formula (8) can always be used to evaluate whether the candidate feature can be finally selected; specifically, formula (8) can be used as an evaluation function. If so, then, obviously, there are two distinct advantages: (1) it is suitable for nominal data; and (2) it can prevent features that have more redundancy from being selected.

To illustrate the above second advantage, we assume an extreme case in which $f_i$ only has more redundancy with one of the features in subset $S$, whereas there is less redundancy or even non-redundancy with the other features. In this case, the value

of $\left(\frac{1}{|S|}\right)\sum_{s\in S}\frac{I(f;s)}{\min\{H(f),H(s)\}}$ is less, and the candidate feature $f_i$ can be certainly selected into the subset $S$. The result enables the subset $S$ to still have more redundancy. To overcome this extreme case, the following countermeasures are proposed. By adjusting the penalty factor in the evaluation function, for example, the second part in Formula (8) can be replaced with a stronger penalty factor, such as the second part of Formula (9).

$$J(f_i) = G(S \cup f_i; C) - \text{argmax}\left\{\frac{G(f_s; f_i)}{\min\{H(f_i), H(f_s)\}}, f_s \in S\right\}. \tag{9}$$

Obviously, this extreme case can be overcome. Based on the above, the Redundancy-removing MRLR **RedremovingMRLR** algorithmis summarized below.

---

**Algorithm: AlgorithmRedremovingMRLR**

Step 1    Initialization: suppose **F** is the universal set with all features, and **S** is the empty set; initialize the value of **k**, which represents the dimensional number of the feature subset that was selected by the feature selection algorithm;

Step 2    Compute the relevance degree according to Formula (3); for each feature $f_i \in F$, compute $G(f_i; C)$;

Step 3    According to the computational results in Step 2, $f_i$ with the maximum relevance degree value $G(f_i; C)$ is selected, and $F \leftarrow F - \{f\}$, $S \leftarrow \{f\}$;

Step 4    For each $f_i$ in the candidate features, compute the preliminary evaluation value by Formula (4). If the preliminary evaluation value of $f_i$ is less than or equal to the average, then compute the evaluation value of $f_i$ according to Formula (8), whereas if the preliminary evaluation value of $f_i$ is greater than the average, then compute the evaluation value of $f_i$ according to Formula (9);

Step 5    Successively, feature $f_i$ with the maximum evaluation value is selected as the next valid feature, and set $F \leftarrow F - \{f_i\}$, $S \leftarrow \{f_i\}$;

Step 6    If $|S| = k$ cannot be satisfied, then turn to Step 4;

Step 7    Output subset $S$.

---

The determining process of $k$ in **RedremovingMRLR** is as follows. At the beginning, the algorithm computes $\frac{R(F)}{|F|}$ on the raw dataset as the initialized number of $k$. While an inflection point appears for the classification accuracy of the **RedremovingMRLR** algorithm on the selected subset S before and after being added to the next candidate feature, **RedremovingMRLR** computes $\frac{R(S)}{|S|}$. As long as $\frac{R(S)}{|S|} \geq \frac{R(F)}{|F|}$ is satisfied, the norm $|S|$ is the value of $k$. Specifically, $k = |S|$, and the $|S| = k$ is the stopping condition of **RedremovingMRLR**. Here, $R(X)$ is the classification accuracy of a certain employed classifier on a certain selected subset or dataset $X$.

The time cost of the **RedremovingMRLR** algorithm contains mainly two parts. One is the time to compute the relevance degree between the features; its time complexity can be noted as $mn\log_2 n$. The other is the time to obtain the final $k$ of the subset **S**, which requires $k$ iterations. Thus, its time complexity is $kmn\log_2 n$, and the total time complexity of **RedremovingMRLR** is $O(mn\log_2 n)$. At this point, the time complexity is the same as that in MIFS and MIFS-U, which clearly shows that the **RedremovingMRLR** algorithm realizes the nominal-data feature selection without increasing the time complexity.

However, the **RedremovingMRLR** algorithm is not always the perfect approach. If the extreme feature $f_i$ is selected as the first feature in subset **S**, then it is inept at obtaining a

**Table 1** The experimental benchmarking nominal datasets are employed in this manuscript.

| Data set | Number of features | Number of patterns | Classes |
|----------|--------------------|--------------------|---------|
| Soybean | 35 | 307 | 19 |
| Vehicle | 18 | 946 | 4 |
| krvs | 36 | 3,196 | 2 |

prime result. Basically, it can be seen that the new evaluation function in the algorithm has strong application flexibility.

The RedremovingMRLR algorithm also employs the methods in MIFS (*Battiti, 1994*) and Reference (*Hu, Xie & Yu, 2007*) to compute $H(f_i)$, $H(f_s)$ in related formulas. Although the probability-equaling discretization processing is conducted on the continuous features, by summing the information entropy of each feature after discretization, MIFS (*Battiti, 1994*) and Reference (*Hu, Xie & Yu, 2007*) mainly focused on the feature selection of continuous-attributed data. Because the nominal data are discrete, it is feasible to equivalently replace $H(f_i)$, $H(f_s)$ in formulas that have nominal features and to directly compute the information entropy. Therefore, the formulas can be directly taken as the evaluation methods with respect to the redundancy between the candidate features and the selected nominal-data features.

# RESULTS AND DISCUSSION

## Experimental data sets

In this section, experiments are performed on 3 benchmarking nominal datasets (Table 1) from the UCI machine learning repository (*Blake & Merz, 2013*). In each problem, all of the patterns that have missing feature values are initially removed. The dataset king-rook vs. king-pawn is represented as krvs briefly.

## Experimental results

In this section, we arrange two experiments to test the feature selection performance, removing the redundancy capability and robust capability of RedremovingMRLR for nominal data. The decision tree classifier (*Brodley & Utgoff, 1995*) and the naive Bayes classifier (*Liu, 2003*) are employed to evaluate the nominal data subsets.

A comparative study between different algorithms is also performed in terms of the indexes for three aspects, namely, (1) the number of final selected features; (2) the classification accuracy of the selected feature subset using different employed classifiers; and (3) the performance of the classification model that is created for different classifiers and both the feature selection complexity and feature classification accuracy. Based on the selected feature subset using the compared algorithms, candidate features are added one by one until the entire raw data set is addressed, and the classification experiment is conducted to evaluate the performance of the algorithm. For NMIFS, the best parameter is selected.

**Table 2 The selected Subsets by Redundancy-removing MRLR, MRLR and NMIFS algorithm respectively.**

| Dataset | FullSet | MRLR | NMIFS | RedremovingMRLR |
|---------|---------|------|-------|-----------------|
| Soybean | 35 | 28 | 24 | 18 |
| Vehicle | 18 | 13 | 16 | 14 |
| Krvs | 36 | 10 | 9 | 8 |

**Table 3 The classification accuracy on the different subsets from the Table 1 by the decision tree classifier.**

| Dataset | NMIFS (%) | MRLR (%) | Redremoving MRLR (%) |
|---------|-----------|----------|----------------------|
| Soybean | $90.4255 \pm 0.0835$ | $91.1032 \pm 0.1132$ | $92.0213 \pm 0.0653$ |
| Vehicle | $70.8034 \pm 0.0584$ | $74.1135 \pm 0.0452$ | $74.3499 \pm 0.0621$ |
| Krvs | $99.2028 \pm 0.1441$ | $99.0257 \pm 0.2167$ | $99.2028 \pm 0.1366$ |

**Table 4 The classification accuracy on the different subsets from the Table 1 by the naive Bayes classifier.**

| Dataset | NMIFS (%) | MRLR (%) | Redremoving MRLR (%) |
|---------|-----------|----------|----------------------|
| Soybean | $91.1348 \pm 0.1567$ | $91.8149 \pm 0.0203$ | $90.9574 \pm 0.245$ |
| Vehicle | $45.7447 \pm 0.0326$ | $48.5814 \pm 0.1095$ | $45.9811 \pm 0.0109$ |
| Krvs | $89.2826 \pm 0.2912$ | $93.4455 \pm 0.0712$ | $95.9256 \pm 0.296$ |

***Experiment 1.*** This experiment aims to quantitatively evaluate the applicability and effectiveness of RedremovingMRLR, and a comparative study among RedremovingMRLR, MRLR and the classical NMIFS is also performed. Here, several high-dimensional datasets (Table 1) that have redundancy features (*Chow, Wang & Ma, 2008*; *Tang & Mao, 2007*; *Li, Yang & Gu, 2013*; *Chert & Yang, 2009*) in UCI (*Blake & Merz, 2013*) are employed to evaluate the feature selection capability of the RedremovingMRLR, MRLR and NMIFS algorithm as well as the naive Bayes classifier and decision tree classifier, which are used to test the effectiveness of the selected subset from the different algorithms. This experiment illustrates the performance of RedremovingMRLR, MRLR and NMIFS for redundant-featured datasets. The experimental results are listed in Tables 2, 3 and 4.

Table 2 lists the final selected feature subsets from the RedremovingMRLR, MRLR and NMIFS algorithms. The results show that the RedremovingMRLR, NMIFS and MRLR algorithms have higher feature-selecting capabilities for high-dimensional, redundant-featured datasets. In the results, we obtain nine new subsets, $Y_i, Y_i \subset \{Y_1, Y_2, \ldots, Y_9\}, i \in [1, 9]$, which correspond to the RedremovingMRLR, MRLR and NMIFS algorithms, respectively. Here, we call them basic feature selected subsets to describe in context. However, on the vehicle dataset for the feature selection by the RedremovingMRLR algorithm, the final subset is one dimension more than that from the

MRLR algorithm. However, on the whole, Table 2 actually illustrates that the performance of RedremovingMRLR is superior to MRLR and NMIFS in terms of its feature-selecting capability.

Table 3 lists the final classification accuracies that were obtained by the decision tree classifier on the nine basic feature selected subsets above. Table 4 lists the final classification accuracies from the naive Bayes classifier on the nine basic feature selected subsets above. For Tables 3 and 4, to make the comparative study fair, an average value over 10 sets of classification results is taken as the estimation value of the classification accuracy, and the 10-fold cross-validation method on them is taken as well as repeated 3 times.

From Table 3, for the decision tree classifier, the classification accuracy of the RedremovingMRLR algorithm on the different basic subsets is the best among the three compared algorithms. RedremovingMRLR demonstrates its advantage over the MRLR and NMIFS algorithms in terms of its feature-selecting capability and effectiveness. Furthermore, the results show that the decision tree classifier is appropriate for classifying the nominal data. For the naive Bayes classifier, the classification accuracy of RedremovingMRLR on the krvs's subset is higher than that of the other two compared algorithms. The main reason is that there exists not only redundancy but strong relevance between the features of the krvs dataset (*Tang & Mao, 2005*; *Tang & Mao, 2007*). Additionally, the classification accuracy of RedremovingMRLR on the soybean subset is lower than that of the other two compared algorithms, whereas for the vehicle, it is higher than that of the NMIFS algorithm and lower than that of the MRLR algorithm.

From the analysis, we find that the main reason for these distinct differences is the classifying principle that is implemented in the classifiers, namely, the different theorems that form the bases for the different classifiers. The decision tree classifier primarily operates through the selection of classification features based on the acquired information, such as selecting one or a couple of key features as the rooting key node randomly; then, it classifies the data items into different classes along the tree clues that exist in the dataset. Moreover, each feature in the selected subsets is absolutely independent, and which one acts as the rooted node in the tree should not affect the final classification results. The more independent the selected features are, the less influence they have on the classification results.

On the other hand, the naive Bayes classifier classifies the data items according to the different probability densities of the values of the different features. However, before or after the features are selected, the probability density of the features is varied not only in the raw dataset but also in the selected subset. Thus, the classification results are diverse. As a result, the difference in the results is due to the different classifying principles in the different classifiers rather than in the classification mechanism.

Therefore, the results show that two cases are involved: (1) the specificity of the non-matrix characteristic, disorderliness characteristic, and disparity characteristic in the nominal data; and (2) the decision tree classifier is more suitable for classifying the nominal data.

*Experiment 2.* This experiment aims to evaluate the efficiency and robust capability of RedremovingMRLR, and a comparative study among RedremovingMRLR, MRLR and the classical NMIFS is also performed. The same datasets as in Experiment 1, i.e., $Y_i, i \in [1, 9]$, are the subject of experimentation.

Here, the experimental examples of each time are generated from one of the basic feature-selected subsets (in *Experiment 1*) by adding up one-dimensional features one by one individually, according to the descending order of the values of the evaluation function found in formulas (8) or (9). In this case, a series of new temporary subsets are formed after one of the features is added at each time. To describe clearly the context, we call them temporary subsets. For each temporary subset, the decision tree classifier and Naive Bayes classifier are employed to train, and thus, the two classification results are obtained at each step until all of the features add up completely. The experimental results are shown in Figs. 1–3. Furthermore, to make the comparative study fair, the average value of 10 instances of classification results is taken as the estimated value for the classification accuracy, and the 10-fold cross-validation method on each temporary subset for each algorithm is adopted.

Figures 1A and 1B shows the classification results on the soybean dataset by the decision tree classifier and the Naive Bayes classifier, respectively. The decision tree classifier and the Naive Bayes classifier all approximately achieve the highest classification accuracy at the beginning of the basic selected subset without adding any other features. From these figures, we can readily see that RedremovingMRLR has the best classification accuracies in most of the cases and is comparatively competitive to NMIFS and/or MRLR even when RedremovingMRLR cannot achieve the best classification results. The classification accuracy of RedremovingMRLR always retains stability. This fact indeed indicates the advantage of RedremovingMRLR over NMIFS and MRLR in its robust capability, in an average sense, and shows that RedremovingMRLR has the resistance capability for outer interference.

Figure 2A shows the generalization accuracy of the decision tree classifier that uses as inputs the temporary subsets of features selected by RedremovingMRLR, NMIFS, MRLR. It can be seen that the best results are obtained with RedremovingMRLR for 14 or more features. Each algorithm achieved its best classification accuracy near the number of the basic selected subsets. RedremovingMRLR outperforms NMIFS and MRLR for any number of features. Figure 2B shows that the generalization accuracy of the Naive Bayes classifier for the RedremovingMRLR, NMIFS, MRLR algorithms. MRLR outperforms NMIFS and RedremovingMRLR for any number of features. Obviously, Fig. 2 demonstrates the whole change curves on the vehicle dataset for three algorithms with the two classifiers. It can be easily seen that the variation tendency of the classification accuracy is the same or similar. After reaching the full set, the classification accuracy of the different classifiers is the same. From these experimental results, we find that (1) RedremovingMRLR, MRLR, NMIFS all have redundancy-distinguishing capabilities; (2) the theorem basis of the classifiers certainly affects the classification results, but the affected level is limited; and (3) little interdependency exists in the vehicle dataset besides the redundancy.

Figure 3 shows the best results that are obtained for the number of basic feature selected subsets with each compared algorithm. For the two classifiers, the classification results

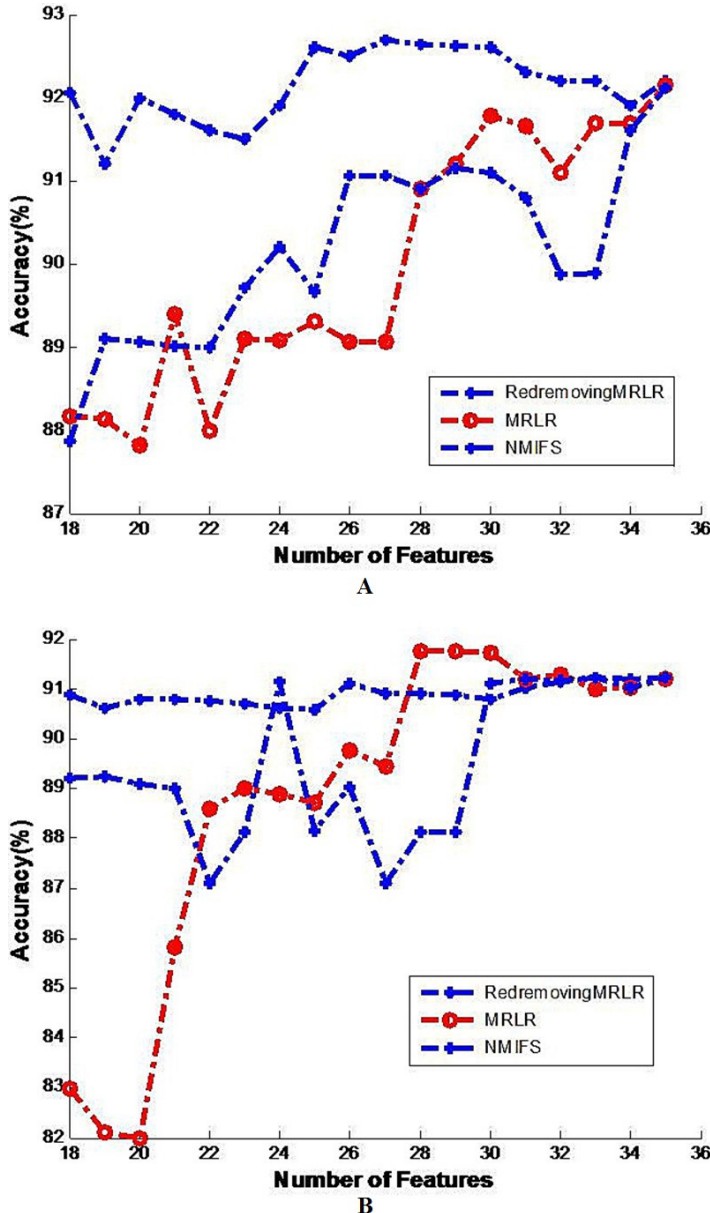

**Figure 1** **Image of generalization classifier accuracy on soybean by different classifiers.** The generalization classifier accuracy on soybean by different classifiers. (A) Shows the accuracy results on soybean by decision tree classifier; (B) Shows the accuracy results on soybean by naive Bayes classifier.

using the 8 features selected by RedremovingMRLR are even better than those using the entire dataset. Figure 3A shows that the changing trend of the classification accuracy by RedremovingMRLR is similar to that of the MRLR algorithm, and it retains a relative height classification rate. Although NMIFS has a better behavior for less than ten features at the beginning, it also has relatively better classification rates after the ten features. In the interval from 10 to 20 features, while selecting some irrelevant or redundant features earlier than the relevant ones, the NMIFS has a bad performance. Figure 3B shows that the

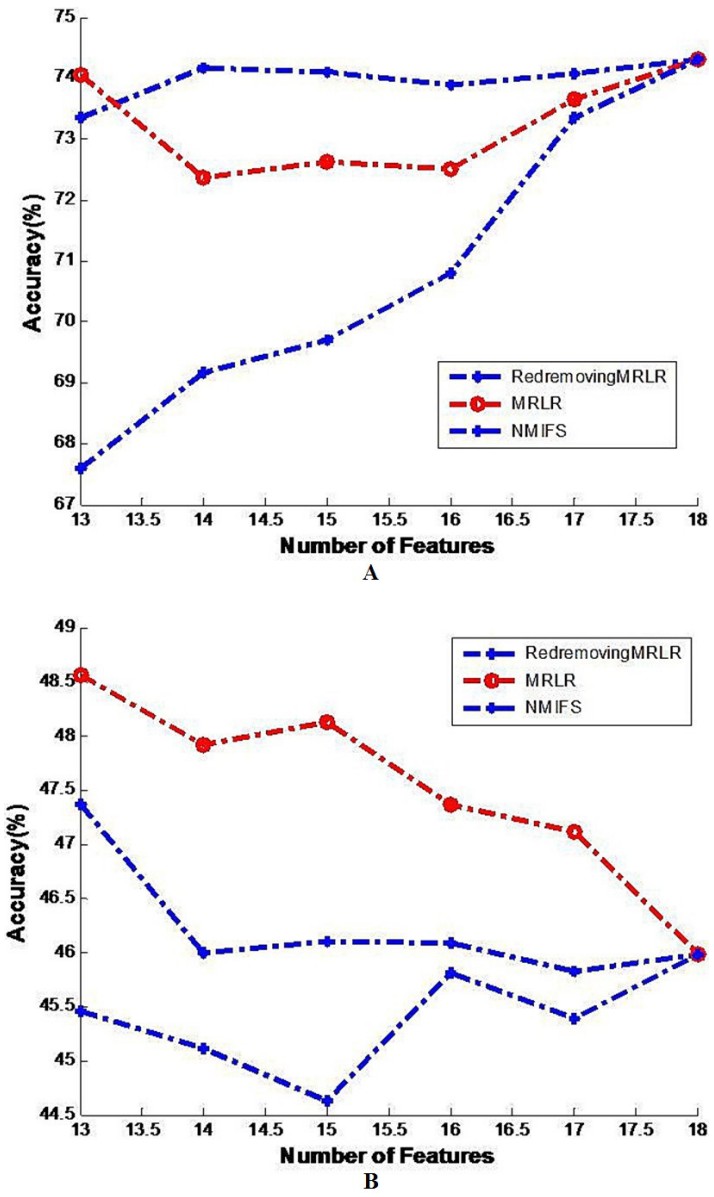

**Figure 2 Image of the generalization classifier accuracy on vehicle by different classifiers.** The generalization classifier accuracy on vehicle by different classifiers. (A) Shows the accuracy results on vehicle by decision tree classifier; (B) Shows the accuracy results on vehicle by naive Bayes classifier.

variation trend of the classification accuracy by the three compared algorithms is similar. On the whole, they are in the desired descending order. After 14 features, the classification accuracy by the NMIFS algorithm is far lower than that of the other compared algorithms; this situation indicates that the performance of NMIFS is influenced by the selection order of the relevant features, redundant features and irrelevant features.

On the whole, from Tables 2, 3 and 4 and Figs. 1–3, besides some wavering variation that exists in Fig. 3B on the krvs dataset by the Naive Bayes classifier for the RedremovingMRLR algorithm, RedremovingMRLR outperforms MRLR and NMIFS with and without

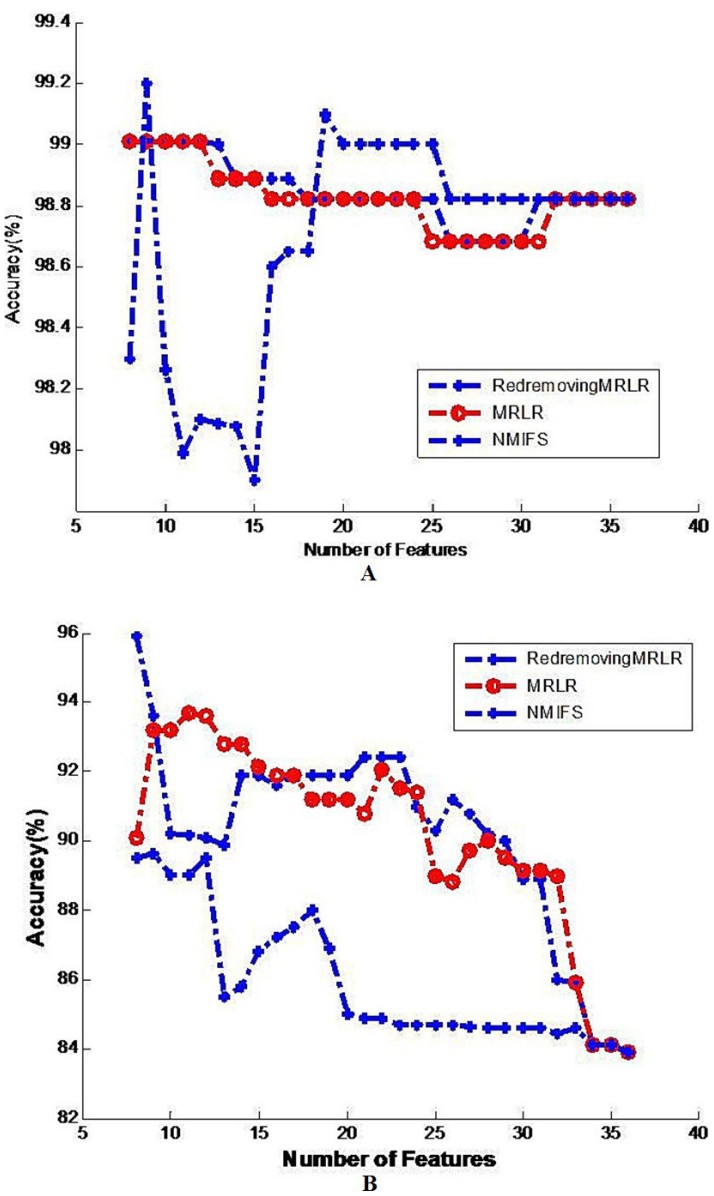

**Figure 3** **Image of the generalization classifier accuracy on krvs by different classifiers.** The generalization classifier accuracy on krvs by different classifiers. (A) Shows the accuracy results on krvs by decision tree classifier; (B) Shows the accuracy results on krvs by naive Bayes classifier.

mutations, finding the best solution with a smaller number of features. The classification accuracy using the 8 features that are selected by RedremovingMRLR are even better than those using the entire dataset. From these experimental results, we find that (1) RedremovingMRLR always selects all of the features that are in the ideal selection order: first, the relevant features in the desired descending order, second, the redundant features, and last, the irrelevant features, rather than using the converse order. In some krvs-like datasets, both MRLR and NMIFS selected some irrelevant features earlier than the redundant features because they penalize too much the redundancy; (2) the experimental results

here indicate that RedremovingMRLR can be applied effectively to nominal data sets with high-dimensional features and has a relatively stronger redundancy-recognizing capability; and (3) the feature selection strategy utilized in RedremovingMRLR is practicability, and the redundancy matrix operator is expressed as formula (7) and its modification in formula (8) and formula (9) make it robust.

## CONCLUSIONS

In this paper, the novel algorithm RedremovingMRLR, which is a method that aims to select features for nominal data, is proposed. The virtues of the proposed algorithm can be summarized as follows: (1) By forming several new information-related definitions for nominal data, such as the information amount, conditional mutual information, and relevance degree, a series of corresponding improvements in their computation methods are presented. With these, the RedremovingMRLR algorithm takes commonly used MIFS-like forms, which enhances its feature selection performance and effectiveness; (2) A reasonable evaluation function of feature selection deems the proposed algorithms to be fit to select the features from the nominal data. However, the computational complexity does not increase, and the feature selection for the nominal data becomes easier; (3) By considering the relevance and redundancy globally and rewriting the evaluation function that is in NMIFS (*Estévez, Tesmer & Perez, 2009*) and then employed by RedremovingM-RLR, its redundancy-removing capability and robust capability are enhanced; (4) Our experimental results demonstrate the average advantage of RedremovingMRLR over the algorithms MRLR and NMIFS in terms of the size of the feature selection subset, the feature efficiency and the classification accuracy.

Improvements on these proposed methods will require further study. An estimation method for MI for nominal data should be developed in the future rather than employing methods from others. The feature selection capability for nominal data with noisy and mixed features as well as the improvement on the corresponding algorithms will be investigated in succeeding studies.

### Funding
This work is supported by the Future Research Projects Funds for the Science and Technology Department of Jiangsu Province (Grant No. BY2013015-23) and the Fundamental Research Funds for the Ministry of Education (Grant No. JUSRP211A 41).The funders had no role in study design, data collection and analysis, decision to publish, or preparation of the manuscript.

### Grant Disclosures
The following grant information was disclosed by the authors:
Science and Technology Department of Jiangsu Province: BY2013015-23.
Fundamental Research Funds for the Ministry of Education: JUSRP211A 41.

## Competing Interests

The authors declare there are no competing interests.

## Author Contributions

- Zhihua Li contributed reagents/materials/analysis tools, wrote the paper, reviewed drafts of the paper.
- Wenqu Gu conceived and designed the experiments, performed the experiments, analyzed the data, prepared figures and/or tables, performed the computation work.

## Data Availability

UCI repository of machine learning database

http://www.ics.uci.edu/∼mlearn/MLRepository.

## Supplemental Information

Supplemental information for this article can be found online at http://dx.doi.org/10.7717/peerj-cs.24#supplemental-information.

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
