# Peer review of "A redundancy-removing feature selection algorithm for nominal data"

_PeerJ Computer Science, doi:10.7717/peerj-cs.24_

## Round 0.1 · original submission · Major Revisions

Both reviewers are positive to your work, please carefully revise your article according to the raised comments.

Reviewer 1 ·

Basic reporting

No Comments

Experimental design

No Comments

Validity of the findings

No Comments

Additional comments

In this paper, the author proposed an algorithm named as RedremovingMRLR to select features for nominal data. My main concerns include:
(1) I think the novelty of this algorithm is the improvement of traditional NMIFS. Please highlight your unique contributions.
(2) In experiments, the computational time of the algorithms should also be compared. The author should analyze the computational complexity at least.
(3) There are some typos in this paper. For example Ref. [16]. Besides, there are also some related works should be cited. For example, for feature extraction, “Stable local dimensionality reduction approaches” and “Multiple view semi supervised dimensionality reduction” are two important literatures. For feature selection, I suggest the author to refer “Joint embedding learning and sparse regression: A framework for unsupervised feature selection”

Reviewer 2 ·

Basic reporting

No Comments.

Experimental design

No Comments.

Validity of the findings

No Comments.

Additional comments

This paper proposes a nominal-data feature selection method based on mutual information . The writting of the paper is very clear. The paper is well organized, and is very easy to follow. The idea in the paper is nice, and it could be accepted with minor revision.

Minor comments:
1. The definition of || in Eq.1 and Eq.4 should be clarified and the * in Eq.2 should be deleted.
2 The font size of data set in table 1should be the same.
3 The size of ...(Y_1, Y_2, ... Y_9) used in Experiment 1.
4 Fig. 1a, 1b shows the classification results .... should be Fig. 1a, and Fig. 1b. show (page 11).

---

## Round 0.2 · accepted · Accept

All comments are addressed and it can be accepted now.

Reviewer 1 ·

Basic reporting

No Comments

Experimental design

No Comments

Validity of the findings

No Comments

Additional comments

The authors have addressed all my comments and I think it is ready for publication.

Reviewer 2 ·

Basic reporting

No Comments

Experimental design

No Comments

Validity of the findings

No Comments

Additional comments

No Comments